# Antioxidant and Anti-α-Glucosidase Activities of Various Solvent Extracts and Major Bioactive Components from the Fruits of *Crataegus pinnatifida*

**DOI:** 10.3390/antiox11020320

**Published:** 2022-02-06

**Authors:** Yen-Ting Lin, Hsiang-Ru Lin, Chang-Syun Yang, Chia-Ching Liaw, Ping-Jyun Sung, Yueh-Hsiung Kuo, Ming-Jen Cheng, Jih-Jung Chen

**Affiliations:** 1Department of Pharmacy, School of Pharmaceutical Sciences, National Yang Ming Chiao Tung University, Taipei 112, Taiwan; ytl97.ps09@nycu.edu.tw (Y.-T.L.); tim0619@nycu.edu.tw (C.-S.Y.); 2Department of Chemistry, College of Science, National Kaohsiung Normal University, Kaohsiung 824, Taiwan; t3136@nknu.edu.tw; 3National Research Institute of Chinese Medicine, Ministry of Health and Welfare, Taipei 112, Taiwan; liawcc@nricm.edu.tw; 4Department of Biochemical Science and Technology, National Chiayi University, Chiayi 600, Taiwan; 5National Museum of Marine Biology and Aquarium, Pingtung 944, Taiwan; pjsung@nmmba.gov.tw; 6Department of Medical Research, China Medical University Hospital, China Medical University, Taichung 404, Taiwan; kuoyh@mail.cmu.edu.tw; 7Department of Biotechnology, Asia University, Taichung 413, Taiwan; 8Department of Chinese Pharmaceutical Sciences and Chinese Medicine Resources, College of Pharmacy, China Medical University, Taichung 404, Taiwan; 9Bioresource Collection and Research Center (BCRC), Food Industry Research and Development Institute (FIRDI), Hsinchu 300, Taiwan

**Keywords:** *Crataegus pinnatifida*, various solvent extracts, active components, antioxidant activity, anti-α-glucosidase activity, molecular docking

## Abstract

*Crataegus pinnatifida* is used to treat various diseases, including indigestion, congestive heart failure, hypertension, atherosclerosis, and myocardial dysfunction. We evaluated antioxidant and anti-α-glucosidase activities of various solvent extracts and major bioactive components from the fruit of *C. pinnatifida*. Ethyl acetate extracts showed potent antioxidant activities with IC_50_ values of 23.26 ± 1.97 and 50.73 ± 8.03 μg/mL, respectively, in DPPH and ABTS radical scavenging assays. Acetone extract exhibited significant anti-α-glucosidase activity with IC_50_ values of 42.35 ± 2.48 μg/mL. HPLC analysis was used to examine and compare the content of active components in various solvent extracts. We isolated four active compounds and evaluated their antioxidant and anti-α-glucosidase properties. Among the isolated compounds, chlorogenic acid and hyperoside showed potential antioxidant activities in ABTS and superoxide radical scavenging assays. Moreover, hyperoside also displayed stronger anti-α-glucosidase activity than other isolates. The molecular docking model and the hydrophilic interactive mode of anti-α-glucosidase assay revealed that hyperoside might have a higher antagonistic effect than positive control acarbose. The present study suggests that *C. pinnatifida* and its active extracts and components are worth further investigation and might be expectantly developed as the candidates for the treatment or prevention of oxidative stress-related diseases and hyperglycemia.

## 1. Introduction

Free radicals are generated from normal biochemical reactions in the body and increase oxidative stress and probably damage biological molecules such as lipids, proteins, and DNA [1]. Free radicals are associated with cancer, atherosclerosis, aging, inflammation, ischemic heart disease, diabetes, and neurodegenerative disorders [2]. To diminish the oxidative stress of free radicals, many synthetic antioxidants were used in industries to delay or inhibit cellular damage mainly through their free radical scavenging property [3]. Natural antioxidants from plant products received attention since they may be safer and more effective in reducing reactive oxygen species (ROS) levels compared to synthetic single dietary antioxidants [4].

Phenolic compounds are free radical scavengers (FRS) that delay or repress the initiation step or hinder the propagation step of lipid oxidation [5]. The antioxidant mechanism of phenolic compounds is that the aromatic ring donates H^+^ to the free radicals during oxidation and become a radical themselves. These radical intermediates are stabilized by the resonance delocalization of the electron within the aromatic ring [6]. For the excellent antioxidant capacity of phenolic compounds, they are regarded as natural antioxidants.

Oxidative DNA damage links pathogenically to multiple aging-regarding degenerative diseases such as cancer, coronary heart disease, and diabetes [7]. Growing evidence of both experimental and clinical studies suggests that there is an affinity between hyperglycemia, oxidative stress, and diabetic complications [8]. ROSs result in defective insulin gene expression and insulin secretion as well as increased apoptosis to increase the risk of DM [9].

α-Glucosidase is an essential enzyme in human carbohydrate metabolism. Many studies have affirmed the association between α-glucosidase inhibitors and blood glucose regulation in type II diabetic patients [10]. α-Glucosidase enzymes are in the brush-border surface membrane of intestinal cells and catalyze the hydrolysis of the α-glycosidic bond of oligosaccharides to release monosaccharide units from dietary sources. Thus, α-glucosidase inhibitors can delay glucose assimilation and decrease postprandial plasma glucose levels [11]. α-Glucosidase inhibitors such as voglibose, acarbose, and miglitol can control postprandial hyperglycemia in DM patients. Nevertheless, these drugs demonstrate adverse side effects, including hepatotoxicity, weight gain, and cardiovascular problems [10].

*Crataegus pinnatifida* is a traditional Chinese medicinal (TCM) herb that is classified as the Rosaceae family and is widely distributed in the north of China [12]. Fruits of *C. pinnatifida* can be used as medicine and food for its safety [13]. In traditional Chinese medicine, *C. pinnatifida* is used in prescriptions to treat indigestion with epigastric distention, abdominal pain, diarrhea, hyperlipidemia, amenorrhea, and hypertension [14]. Previous studies have demonstrated that *C. pinnatifida* also possesses various biological activities, including lipid regulating, anti-atherosclerosis, antihypertensive, antibacterial, antiviral, anticancer, immunoregulating, anti-inflammatory, and antioxidant activities [12,15,16,17,18]. In the present study, we investigated the antioxidant and anti-α-glucosidase effects of different solvent extracts and major bioactive components from the fruits of *C. pinnatifida*. Moreover, we conducted molecular docking between α-glucosidase and hyperoside with the most potent anti-α-glucosidase activity. In addition, the amounts of isolated compounds were quantified by HPLC analysis. The antioxidant and anti-α-glucosidase effects of isolated components were also evaluated.

## 2. Materials and Methods

### 2.1. Chemicals and Reagent

Ascorbic acid, 2,2’-azino-bis(3-ethylbenzothiazoline-6-sulfonic acid) (ABTS), α-glucosidase, ethylenediaminetetraacetic acid (EDTA), gallic acid, Folin-ciocalteau’s reagent, hydrogen peroxide solution, Trolox, and 2,4,6-Tris(2-pyridyl)-s-triazine (TPTZ) were purchased from Sigma-Aldrich (St. Louis, MO, USA). Ferric chloride (FeCl_3_) and *p*-nitro-phenyl-α-D-glucopyranoside (*p*-NPG) were obtained from Alfa Aesar (Lancashire, UK). Disodium hydrogenphosphate, potassium peroxodisulfate, sodium carbonate, and potassium dihydrogenphosphate were acquired from the SHOWA Chemical Co. Ltd. (Chuo-ku, Japan). 2,2-diphenyl-1-(2,4,6-trinitrophenyl)hydrazyl (DPPH), deoxyribose, nitroblue tetrazolium (NBT), phenazine methosulphate (PMS), and 2-thiobarbituric acid (TBA) were supplied by Tokyo Chemical Industry Co., Ltd. (Tokyo, Japan). Butyl hydroxytoluene (BHT), acarbose, nicotinamide adenine dinucleotide (NADH), sodium acetate, trichloroacetic acid (TCA), and potassium acetate were acquired from Acros Organics (Geel, Belgium). Acetic acid was supplied by Avantor Performance Materials, LLC (Radnor, PA, USA). Sodium hydroxide solution was acquired from Merck (Darmstadt, Germany).

### 2.2. Preparation of C. pinnatifida Extract

The fruits of *C. pinnatifida* were collected from Wanhua Dist., Taipei City, Taiwan, in July 2020 and identified by Prof. J.-J. Chen. A voucher specimen was deposited in the Department of Pharmacy, National Yang Ming Chiao Tung University, Taipei, Taiwan. Samples were obtained, air-dried, and cut into pieces. The amount of 125 mL of different solvents (*n*-hexane, chloroform, dichloromethane, EtOAc, acetone, EtOH, and MeOH) was added into 20 g of pieces and shaken by orbital shakers for 24 h at 25 °C. The extracts were filtered through filter paper (Whatman No. 1) and condensed by a rotary evaporator at 37 °C. All extracts were stored at −20 °C until further experiments.

### 2.3. Preparation of Active Components

The fruits (20 g) of *C. pinnatifida* were extracted and shredded with MeOH (3 × 125 mL, 3 d each) at room temperature. The MeOH extract was concentrated under reduced pressure at 37 °C, and the MeOH extract (5.61 g) was obtained. MeOH extract (5.61 g) was purified by column chromatography (CC) (220 g of silica gel, 70–230 mesh; *n*-hexane/acetone gradient) to afford 14 fractions: A1–A14. Part (153 mg) of fraction A9 was further purified by preparative TLC (silica gel; *n*-hexane/acetone, 2:1) to afford epicatechin (4.98 mg) (R_f_ = 0.12) and procyanidin B2 (3.81 mg) (R_f_ = 0.35). Part (215 mg) of fraction A11 was further by preparative TLC (silica gel; dichloromethane/MeOH, 5:1) to obtain hyperoside (1.70 mg) (R_f_ = 0.30) and chlorogenic acid (5.81 mg) (R_f_ = 0.23). The TLC graphs of isolated compounds are shown in Appendix A. The structures of epicatechin, chlorogenic acid, hyperoside, and procyanidin B2 were identified by nuclear magnetic resonance (NMR) spectra acquired using a Bruker Avance 400 MHz spectrometer (Bruker, Bremen, Germany).

### 2.4. Reverse-Phase HPLC

Reversed-phase separations were performed using a LiChrospher^®^ 100 RP-18 Endcapped (5 μm; column of dimensions 4.6 × 250 mm) purchased from Merck KGaA, Darmstadt, Germany. HPLC-PDA chromatographic fingerprints were obtained with an Agilent 1260 Infinity II HPLC instrument equipped with a 1260 Infinity II quaternary pump, a 1260 Infinity II degasser, a 1260 Infinity II vialsampler, a 1260 Infinity II column thermostat, a 1260 Infinity II diode array detector HS, and a PC with the Agilent Chemstation software. All of them are from Agilent Technologies (Waldbronn, Germany). Gradient separation using 0.2% acetic acid in water (*v*/*v*) (solvent A) and water:methanol:acetonitrile = 3:2:5 (solvent B) as mobile phase was as follows: 0–20 min, linear gradient from 0 to 15% B; 20–65 min, 15% B with isocratic elution; 65–110 min, linear gradient from 15 to 40% B; 110–120 min, linear gradient from 40 to 100% B; 120–130 min, back to initial conditions at 0% B; and 130–135 min, at 0% B. The flow rate was 1.0 mL/min, and the injection volume was 10 μL. Peaks were detected at 280 nm. Different compounds were identified by retention time. To guarantee peak purity, DAD acquisition from 200 to 650 nm was conducted to register UV-spectra. For the quantitative analysis of four compounds in the extracts, aliquots of samples were dispersed in 10 mL of a methanol solution by sonication for 5 min. Then, the samples were centrifuged for 15 min at 3500 rpm, and the supernatant extracts were filtered through 0.45 μm PTFE syringe filters (Zhejiang Sorfa Medical Plastic Co., Ningbo, China).

### 2.5. Determination of Total Phenolic Content

Total phenolic content (TPC) of different extracts was determined by Folin-Ciocalteau’s method with a slight modification of [19]. Briefly, 0.5 N Folin-Ciocalteu reagent (50 μL) was diluted by ddH_2_O. Gallic acid was diluted by MeOH (0–100 μg/mL) as standard. Each extract or gallic acid (50 μL) was added to a 96-well microplate. The mixture was incubated for 5 min. Subsequently, 20% Na_2_CO_3_ solution (100 μL) was added. After 40 min, it was incubated in darkness at room temperature before measuring the absorbance of the supernatant at 750 nm. The TPC of the extracts was determined from a standard calibration curve using gallic acid in the range of 0–100 μg/mL with R^2^ value of 0.9992. The concentration of TPC was expressed in milligram (mg) gallic acid equivalents (GAE) per gram of dried extract. All measurements were conducted in triplicate.

### 2.6. Determination of Total Flavonoid Content

The total flavonoid content (TFC) of each extract was measured using the aluminum chloride colorimetry method described by Do et al. [20] with slight modifications. In short, the extracted sample was diluted with MeOH to reach a concentration of 100 μg/mL. Quercetin was diluted in MeOH (0–100 μg/mL) as standard. The diluted extract sample or quercetin (400 µL) was mixed with 10% (*w*/*v*) aluminum chloride solution (200 µL) and 0.1 mM potassium acetate solution (200 µL). The mixture was reacted at room temperature for 30 min. Then, the absorbance of the mixture was measured at 415 nm. The TFC of the extracts was determined from a standard calibration curve using quercetin with R^2^ value of 0.9997. The concentration of TFC was expressed in milligram quercetin equivalents (GAE) per gram of dried extract. All measurements were conducted in triplicate.

### 2.7. DPPH Radical Scavenging Activity

The scavenging activity of DPPH radical was investigated as described with slight modification [21]. DPPH solution was prepared diluted by EtOH to reach the final concentration of 200 µM. Different concentrations (200, 100, 50, 25, and 12.5 µg/mL) of extracts or compounds (100 µL) were mixed with DPPH solution (100 µL) and incubated for 30 min at room temperature and in darkness, and the absorbance of the mixture was measured at 520 nm. DPPH radical scavenging activity was calculated with the following formula:
DPPH scavenging activity (%) = (A_c_ − A_t_)/A_c_ × 100,
where A_t_ is the absorbance of the test sample, and A_c_ is the absorbance of the control (untreated group). BHT was used as the positive control. All half-maximal inhibitory concentration (IC_50_) values of tested activities were determined by the linear regression of the percentage of remaining DPPH radical against the sample’s concentration.

### 2.8. ABTS Radical Scavenging Activity

ABTS radical scavenging activity of each extract was measured based on the method with slight changes [22]. In brief, the stock of working solution was prepared by the mix of 28 mM ABTS solution and 9.6 mM potassium persulfate with ddH_2_O (final concentration, 1/1, *v*/*v*) and leaving the mixture in the dark for about 16 h at room temperature before use. The working solution was diluted by EtOH to reach the absorbance of 0.70 ± 0.02 at 740 nm for further experiments. Different concentrations (200, 100, 50, 20, and 12.5 µg/mL) of extracts or compounds (10 µL) were mixed with the working solution (190 μL). The mixture was incubated at room temperature for 6 min, the antioxidant activity of the mixture was determined by calculating the decrease in absorbance measured at 740 nm by the following equation:ABTS radical inhibiting activity (%) = (A_c_ − A_t_)/A_c_ × 100,

Where A_c_ and A_t_ are the absorbance of the control (untreated group) and test sample, respectively. The IC_50_ values of all tested activities were determined by the liner regression of the percentage of remaining ABTS radical against the sample’s concentration.

### 2.9. Superoxide Radical Scavenging Activity

The scavenging activity of superoxide anion radical (O_2_^•−^) was evaluated using a modified method [23]. NBT (300 μM), NADH (468 μM), and PMS (120 μM) were prepared with dilution by Tris-HCl buffer (16 mM, pH 8.0), respectively. An amount of 50 μL of NBT, PMS, and different concentrations (400, 200, 10, 50, and 25 µg/mL) of extracts or compounds were mixed. An amount of 50 μL NADH solution was added, and the reaction was initiated to produce superoxide radicals. The absorbance was measured at 560 nm after incubating at room temperature for 5 min. The scavenging activity was measured by the following equation:Superoxide radical scavenging activity = (A_c_ − A_t_)/A_c_ × 100,
where A_t_ is the absorbance of the test sample, and A_c_ is the absorbance of the control (untreated group). The IC_50_ values of all tested activities were determined by the linear regression of the percentage of remaining superoxide radical against the sample concentration.

### 2.10. Hydroxyl Radical Scavenging Activity

The scavenging activity of hydroxyl radical was measured based on the method described by Mathew and Abraham with slight revision [24]. In short, different concentrations (400, 200, 10, 50, and 25 µg/mL) of extracts or compounds (200 μL) were mixed with working solution (200 μL) contained 1 mM FeCl_3_, 1 mM EDTA, and 2.8 mM deoxyribose. Then, 20 μL ascorbic acid (1 mM) and 100 μL H_2_O_2_ (20 mM) in sodium phosphate buffer (20 mM pH 7.4) was added. The mixture was incubated in a water bath at 37 °C for 1 h to trigger the Fenton reaction and form a hydroxyl radical. After incubation, 2.8% TCA (200 μL) and 1% TBA (200 μL) were added and boiled in the dry bath at 100 °C for 1 h. The mixture was measured at 532 nm by the following equation:Hydroxyl radical scavenging activity (%) = (A_c_ − A_t_)/A_c_ × 100, 
where A_t_ and A_c_ are the absorbance of the test sample and the control (untreated group), respectively. The IC_50_ values of all tested activities were determined by the linear regression of the percentage of remaining hydroxyl radical against the sample’s concentration.

### 2.11. Ferric Reducing Antioxidant Power (FRAP)

Ferric reducing antioxidant power was measured based on previous study with slight modification [25]. The working solution was mixed with acetate buffer (pH 3.6), ferric chloride solution (20 mM), and TPTZ solution (10 mM TPTZ in 40 mM HCl) in a proportion of 10:1:1, respectively, and freshly prepared before used. An amount of 900 μL of the working solution was warmed to 37 °C and then mixed with 100 μL of the diluted sample, blank or standard in a microcentrifuge tube. The tubes were vortexed and in the dry bath at 37 °C for 40 min. Absorbance was measured at 593 nm. The standard curve was linear between 0 and 100 mM Trolox with an R^2^ value of 0.998. Results are expressed in mM TE/g dry weight. Additional dilution was needed if the FRAP value measured was over the linear range of the standard curve.

### 2.12. α-Glucosidase Inhibitory Activity Assay

The inhibition assay of α-glucosidase was conducted using the conditions previously reported with slight modifications [26]. The α-Glucosidase solution was diluted by 0.1 M sodium phosphate buffer (pH 6.8) to 1 U/mL. Different concentrations (400, 200, 10, 50, and 25 µg/mL) of extracts or compounds (100 μL) were mixed with α-glucosidase solution (20 μL) in a microcentrifuge tube. Subsequently, *p*-NPG (0.53 mM; 380 μL), the substrate, was added and incubated in dry baths at 37 °C for 40 min after the mixture vortexed. The reaction was terminated after adding 0.1 M Na_2_CO_3_ solution (500 μL). The absorbance of released *p*-nitrophenol *(p*-PNP) was measured by calculating the decrease in absorbance measured at 400 nm by the following equation:α-Glucosidase inhibition (%) = (A_c_ − A_t_)/A_c_ × 100, 
where A_c_ is the absorbance of the control (untreated group), and A_t_ is the absorbance of the test sample. The IC_50_ values of all tested activities were determined by the linear regression of the percentage of remaining α-glucosidase against the sample concentration.

### 2.13. Molecular Modeling Docking Study

All calculations are performed by Discovery Studio 2019 (San Diego, CA, USA) software. Primarily, the structure of hyperoside or acarbose is energy minimized until the default derivative convergence criterion of 0.01 kcal/mol is met. The crystal structure (PDB: 3A4A) is retrieved from the Protein Databank, and hydrogen atoms are added to prepare the docked receptor. This protein structure is subsequently used in the CDocker program to dock hyperoside or acarbose into the active site. Ten different docking poses are calculated and ranked by using the PLP-Score scoring function. The top-ranked docking solution is visually analyzed to determine the binding mode of hyperoside or acarbose.

### 2.14. Statistical Analysis

All data are expressed as mean ± SEM. Statistical analysis was carried out using Student’s *t*-test. A probability of 0.05 or less was considered statistically significant. All experiments were performed at least 3 times.

## 3. Results and Discussion

### 3.1. Determination of Total Phenolic Content (TPC), Total Flavonoid Content (TFC) and Yields in Each Solvent Extract

TPCs, TFCs, and yields of various solvent extracts from *C. pinnatifida* were evaluated. TPCs, TFCs, and extraction yields of *n*-hexane, chloroform, dichloromethane, EtOAc, acetone, EtOH, and MeOH extracts from *C. pinnatifida* are shown in Table 1.

The yields of various solvent extracts from *C. pinnatifida* ranged from 2.3 ± 1.67% (*n*-hexane extract) to 28.0 ± 0.75% (MeOH extract). Higher polar solvent (including MeOH and EtOH) extracts had higher yields. The results indicated that *C. pinnatifida* is abundant in high polar components.

Phenolic contents in different solvent extracts from *C. pinnatifida* were measured by the Folin–Ciocalteu method. TPC in the extracts was calculated from the regression equation (y = 0.0524x, R^2^ = 0. 9989) of the calibration curve. The TPCs of solvent extracts ranged from 12.12 ± 0.26 to 63.53 ± 0.27 mg of GAE/g. Among all different solvent extracts, the EtOAc extract contained the highest amount of TPC (63.53 ± 0.27 mg/g) followed by acetone (45.81 ± 0.61 mg/g), MeOH (44.20 ± 1.30 mg/g), EtOH (32.82 ± 1.21 mg/g), chloroform (21.09 ± 0.99 mg/g), dichloromethane (19.77 ± 0.13 mg/g), and *n*-hexane (12.12 ± 0.26 mg/g). These results suggested that solvents with higher relative polarity are suitable for the extraction of phenolic compounds from *C. pinnatifida*.

The TFCs of solvent extracts from *C. pinnatifida* were evaluated by aluminum chloride colorimetric method. TFC in the extracts was calculated from the regression equation (y = 0.0459x, R^2^ = 0. 9997) of the calibration curve. The TFCs of different solvent extracts ranged from 17.40 ± 3.11 to 37.93 ± 5.31 mg of QE/g. Among all different solvent extracts, chloroform extract contained highest amount of TFC (37.93 ± 5.31 mg/g) and followed by dichloromethane (34.87 ± 1.95 mg/g), *n*-hexane (31.74 ± 1.07 mg/g), EtOAc (25.22 ± 4.21 mg/g), EtOH (22.61 ± 0.90 mg/g), acetone (19.41 ± 3.00 mg/g), and MeOH (17.40 ± 3.11 mg/g).

The comparative assessment of total phenolic content (TPC) and total flavonoid content (TFC) of various solvent extracts (*n*-hexane, chloroform, dichloromethane, EtOAc, acetone, EtOH, and MeOH) from the fruits of *C. pinnatifida* was first mentioned in this study. This can supply a guide for the option of suitable solvents in TPC and TFC extraction applications.

### 3.2. DPPH Free Radical Scavenging Activity

The hydrogen atom or electron donation abilities of various solvent extracts were measured by the reduction of a purple DPPH^•^ solution to 1,1-diphenyl-2-picryl hydrazine [27,28]. Table 2 showed the DPPH radical scavenging activities of each extract. The EtOAc extract (IC_50_ = 23.26 ± 1.97 μg/mL) exhibited the strongest DPPH radical scavenging activity and was even better than BHT (IC_50_ = 34.28 ± 1.40 μg/mL). In addition, acetone (IC_50_ = 40.06 ± 0.18), chloroform (IC_50_ = 68.61 ± 3.14 μg/mL), dichloromethane (IC_50_ = 76.22 ± 4.90 μg/mL), MeOH (IC_50_ = 74.13 ± 5.61 μg/mL), and EtOH (IC_50_ = 96.21 ± 4.26 μg/mL) extracts showed moderate activities. *n*-Hexane (IC_50_ = 199.18 ± 16.82 μg/mL) extract showed relatively high IC_50_ value.

### 3.3. ABTS Free-Radical Scavenging Activity

The ABTS assay is based on the generation of a blue/green ABTS^•^^+^ that can be reduced by antioxidants. The high-pigmented and hydrophilic antioxidants were better reflected by ABTS assay than DPPH assay [28]. As shown in Table 2, the EtOAc extract also exhibited relatively effective ABTS radical scavenging activity (IC_50_ = 50.73 ± 8.03 μg/mL) followed by acetone (IC_50_ = 87.75 ± 2.70 μg/mL), MeOH (IC_50_ = 105.86 ± 6.25 μg/mL), *n*-hexane (IC_50_ = 131.14 ± 7.15 μg/mL), dichloromethane (IC_50_ = 131.44 ± 7.13 μg/mL), EtOH (IC_50_ = 140.29 ± 6.76), and chloroform (IC_50_ = 141.97 ± 3.70 μg/mL).

### 3.4. Superoxide Radical Scavenging Activity

This assay is measured by the PMS/NADH–NBT system. Superoxide anion radicals generated from dissolved oxygen by PMS/NADH coupling reaction reduce NBT. The decrease in absorbance at 560 nm with antioxidants indicates a reduction in superoxide anion radicals in the reaction mixture [29]. The results are shown in Table 2, and only EtOAc extract exerted a significant effect (IC_50_ = 122.95 ± 9.07 μg/mL) on the superoxide radical scavenging assay.

### 3.5. Hydroxyl Radical Scavenging Activity

This assay evaluated the abilities of the extracts to suppress the degradation of hydroxyl radical bases on a Fenton reaction model system containing FeCl_3_-EDTA-ascorbic acid and H_2_O_2_ [30]. EtOAc extract (IC_50_ = 92.30 ± 7.47 μg/mL) showed relatively potent antioxidant activity in scavenging hydroxyl radical followed by chloroform (IC_50_ = 114.41 ± 1.80 μg/mL), *n*-hexane (IC_50_ = 124.45 ± 6.27 μg/mL), acetone (IC_50_ = 127.25 ± 2.26 μg/mL), MeOH (IC_50_ = 138.04 ± 21.91 μg/mL), and EtOH (IC_50_ = 139.03 ± 17.33 μg/mL) (Table 2).

### 3.6. Ferric Reducing Antioxidant Power

The assay measures the antioxidant potential of each extract through a reduction of ferric iron (Fe^3+^) complex to ferrous iron (Fe^2+^) complex by antioxidants present in the samples [31]. Acetone extract exhibited highest ferric reducing antioxidant powers (264.27 ± 12.11 TE mM/g). In addition, EtOH (227.59 ± 10.13 TE mM/g), EtOAc (216.32 ± 9.13 TE mM/g), and MeOH extracts (198.94 ± 8.24 TE mM/g) showed relatively high reducing ability followed by dichloromethane (154.05 ± 4.62 TE mM/g), chloroform (149.98 ± 5.54 TE mM/g), and *n*-hexane (70.11 ± 4.13 TE mM/g) (Table 2).

Based on the above results of the antioxidant assays, the EtOAc extract from *C. pinnatifida* possessed the highest phenolic content and the highest hydrogen atom or electron donation abilities in DPPH and ABTS radical scavenging assays. In addition, the EtOAc extract also showed potent superoxide and hydroxyl radical scavenging effects. Moreover, acetone extract showed the highest reducing ability in the FRAP assay.

The comparative evaluation of antioxidant assays (DPPH, ABTS, superoxide, hydroxyl, and FRAP) of various solvent extracts (*n*-hexane, chloroform, dichloromethane, EtOAc, acetone, EtOH, and MeOH) from the fruits of *C. pinnatifida* was first proposed in our study. This can provide an indication for the selection of appropriate solvents in antioxidant extraction applications.

### 3.7. Anti-α-Glucosidase Activity Assay

α-Glucosidase inhibitors decrease the rate of hydrolytic cleavage of oligosaccharide and delay carbohydrate digestion [32,33]. The results shown in Table 3, the acetone extract of *C. pinnatifida* exerted the most anti-α-glucosidase activity (IC_50_ = 42.35 ± 2.48 μg/mL) followed by MeOH (IC_50_ = 52.02 ± 0.24 μg/mL), EtOH (IC_50_ = 58.69 ± 6.91 μg/mL), *n*-hexane (IC_50_ = 99.75 ± 4.37 μg/mL), dichloromethane (IC_50_ = 120.41 ± 6.44 μg/mL), EtOAc (IC_50_ = 126.36 ± 9.81 μg/mL), and chloroform (IC_50_ = 207.46 ± 9.52 μg/mL). Seven solvent extracts showed higher anti-α-glucosidase activities than antidiabetic agent, acarbose (IC_50_ = 317.80 ± 16.36). Among all solvent extracts, acetone, MeOH, and EtOH extracts showed relatively potent anti-α-glucosidase activity. These results indicated that the suitable relative polarity of extracting solvents for anti-α-glucosidase activity from *C. pinnatifida* would be ranged from 0.355 to 0.762.

### 3.8. Quantitation of Active Components in Different Solvent Extracts

Appendix A displayed the quantification of active components in different solvent extracts from *C. pinnatifida* by HPLC analysis with reverse-phase. The contents of four active compounds in each solvent extract was shown in Table 4. Total amount of four active compounds in each extract ranged from a maximum of 33.62 ± 0.49 μg/kg (EtOH extract) to a minimum of 1.92 ± 0.25 μg/kg (*n*-hexane extract) in succeeding order of EtOH > MeOH > EtOAc > acetone > dichloromethane > chloroform > *n*-hexane extract. EtOH (33.62 ± 0.49 μg/kg) and MeOH (27.76 ± 0.45 μg/kg) extracts exhibited higher amounts of four active compounds compared with other extracts. Procyanidin B2 was the most abundant among the four active compounds in organic solvent extract followed by chlorogenic acid, epicatechin, and hyperoside.

The comparative evaluation for identification and quantification of the major active components (epicatechin, chlorogenic acid, hyperoside, and procyanidin B2) of different solvent extracts (*n*-hexane, chloroform, dichloromethane, EtOAc, acetone, EtOH, and MeOH) from the fruits of *C. pinnatifida* by HPLC analysis is first conducted in our study.

### 3.9. Antioxidant Activities of Isolated Components

The isolated compounds, epicatechin, chlorogenic acid, hyperoside, and procyanidin B2 (Figure 1), were measured for their antioxidant effects, including ABTS, DPPH, hydroxyl, and superoxide radical scavenging activities. Results are shown in Table 5, four isolated compounds showed potent DPPH and ABTS radical scavenging activities. Hyperoside (IC_50_ = 10.59 ± 0.22 μg/mL) exerted a higher superoxide radical scavenging activity than other isolated compounds. Epicatechin (IC_50_ = 20 ± 3.49 μg/mL), chlorogenic acid (IC_50_ = 18.38 ± 0.56 μg/mL), and procyanidin B2 (IC_50_ = 13.69 ± 1.88 μg/mL) significantly inhibited hydroxyl radical with a lower IC_50_ value than positive control, BHT (IC_50_ = 60.62 ± 2.58 μg/mL). The antioxidant activities of isolates were also evaluated by FRAP assay. Hyperoside, epicatechin, and procyanidin B2 show higher antioxidant power (TE ≥ 7726.67 ± 39.97 mM/g) than BHT (TE = 4257.97 ± 44.90 mM/g).

### 3.10. Anti-α-Glucosidase Activities of Isolated Components

The inhibitory activity of four isolated compounds from *C. pinnatifida* against α-glucosidase is shown in Table 6. The results showed that the inhibitory activities of four isolated compounds against α-glucosidase were stronger than the positive control (acarbose, IC_50_ = 317.80 ± 16.36 μg/mL). Hyperoside and epicatechin displayed significant activities with IC_50_ values of 34.98 ± 0.89 and 81.79 ± 6.94 μg/mL followed by procyanidin B2 (IC_50_ = 118.44 ± 7.34 μg/mL) and chlorogenic acid (IC_50_ = 170.37 ± 10.68 μg/mL).

### 3.11. Molecular Modeling Docking

According to the results of anti-α-glucosidase assay, hyperoside exhibited the most potent anti-α-glucosidase activity among all isolated compounds. Thus, the interaction between α-glucosidase and hyperoside was evaluated by molecular modeling docking. The 3D crystal structure of α-glucosidase showed that it mainly contains numerous structural domains including the N-terminal domain, the barrel domain in which the active site is located, and the C-terminal domain. The α-glucosidase active site is primarily formed by numerous β-sheets and several loops or α-helices. Importantly, the active site from different species shows certain conformational similarities. It mainly contains hydrophilic residues and can accept compounds with distinct sizes to enter. The crystal structure of α-glucosidase from beta vulgaris complexed with acarbose (Figure 2a; PDB: 3W37) reveals that its active site is not deep but wide enough to allow acarbose to enter in the flat conformation [34]. Due to containing four glucosyl-like groups, acarbose mainly interacts with the active site by significant H-bond interactions including (1) the 2-hydroxyl group on the A ring interacting with Arg 552 by acting as the H-bond donor; (2) the amino linker between A and B rings makes H-bond contact with Asp 568 and Arg 552; (3) on the B ring, the 3′-hydroxyl group interacts with Arg 552 by acting as the H-bond acceptor and the 2′-hydroxyl group makes hydrophilic contact with Asp 232 by acting as the H-bond donor; (4) the oxygen atom on the ether moiety of the D ring makes H-bond interactions with Asn 237.

To further study how hyperoside (Figure 2b) might interact with α-glucosidase of *Saccharomyces cerevisiae* to exhibit its antagonistic effect, the docking models of hyperoside were generated by Discovery Studio 2019 (Accelrys, San Diego, USA) CDocker modeling program. The 3D crystal structure for α-glucosidase of *Saccharomyces cerevisiae* is not available at this moment; thus, the crystal structure (PDB: 3A4A) of *Saccharomyces cerevisiae* (PDB: 3A4A) containing 72% sequence homology with α-glucosidase from *Saccharomyces cerevisiae* is usually used to perform the docking study and is also employed in this study. In the crystal structure (PDB: 3A4A), the configuration of its active site is quite similar to that of α-glucosidase from beta vulgaris, but it is deep and narrow. The crystal structure of this active site reveals that its co-crystallized ligand, α-D-glucopyranose, locates deeply in the ligand-binding pocket and makes three essential H-bond interactions including (1) the 4-hydroxyl group interacting with His 351 and Asp 352 by acting as the H-bond donor; (2) the 5-hydroxyl group that makes H-bond contact with Glu 277 as well as Asp 352 by acting as the H-bond donor and also serves as the H-bond acceptor to interact with Arg 213; and (3) the 6-hydroxymethyl group that acts as the H-bond donor to interact with Glu 277 and Asp 352.

As shown in Figure 3a, the docking model of hyperoside indicated that hyperoside did not enter the active site in the flat conformation due to the narrow entrance. Alternatively, the A ring of hyperoside leaned toward the binding position of α-D-glucopyranose and the D ring of hyperoside located nearby the entrance of the active site. Importantly, hyperoside made some significant hydrophilic interactions, including the following: (1) the 5-hydroxyl and 7-hydroxyl groups on the A ring both served as the H-bond donors to interact with Asp 69 and Asp 215, respectively; (2) the 4-keto group on the B ring behaved as the H-bond acceptor to make contact with Arg 442; (3) both 3′-hydroxyl and 4′-hydroxyl groups on the A ring acted as the H-bond donors to interact with His 280 and Asp 307 respectively; and (4) the 5′′-hydroxyl group on the D ring acted as the H-bond donor to interact with Glu 411 and the 6′′-hydroxymehtyl group also on the D ring behaved as the H-bond acceptor to contact with the amido side chain of Asn 415. Apart from hydrophilic interaction, hyperoside also made important hydrophobic contact: the π–π interaction between its B ring and Tyr 158.

As the positive control, the docking model for acarbose is also generated to compare its binding mode with hyperoside. Acarbose was bound to the catalytic site similarly as hyperoside by leaning its A ring toward the binding position of α-D-glucopyranose and located its D ring to protrude out of the entrance of the active site. Since it contains a lot of hydrophilic moieties, acarbose mainly interacted with the active site by the H-bond interactions, including the following: (1) the 5-hydroxymethyl group on the A ring interacted with Asp 215 by acting as the H-bond donor; (2) the 4-hydroxyl group on the A ring interacted with His 351 as well as Asp 352 by acting as the H-bond donor and also served as the H-bond acceptor to contact with Arg 442; (3) the 5′′-hydroxymethyl group on the C ring interacted with His 280 by acting as the H-bond donor; (4) the 1′′′-hydroxyl group on the D ring made H-bond interaction with Lys 156 as well as Ser 240 by acting as the H-bond acceptor, and it also made H-bond contact with Leu 313 by acting as the H-bond donor; (5) the 2′′′-hydroxyl group on the D ring interacted with the backbone of Pro 312 as the H-bond donor; and (6) the 3′′′-hydroxyl group on the D ring contacted with the side chain of Asp 242 by serving as the H-bond donor.

For the comparison of the binding modes of hyperoside and acarbose, hyperoside can reside in the middle of the substrate binding pocket and its D ring is located at the entrance of the pocket to block the entrance and enhance its inhibitory effect. For acarbose, its A ring can be located in a similar position as the A ring of hyperoside but its D ring partially protrudes out of the substrate-binding pocket. For their hydrophilic interactive binding mode (Figure 3b), the A rings of hyperoside and acarbose exert similar binding modes. The B and C rings of hyperoside not only make significant hydrophilic but also hydrophobic interactions with the residues in the middle of the substrate-binding pocket. The B and C rings of acarbose exert only one hydrophilic interaction and they are surrounded by some hydrophobic residues including Tyr 158, Phe 159, and Phe 178; thus, their numerous hydroxyl moieties might not be able to exhibit significant hydrophobic interactions as the B ring of hyperoside. Moreover, the D ring of hyperoside resides at the entrance of the substrate-binding pocket to block the entrance through two essential H-bond interactions but the D ring of acarbose protrudes out of the pocket and exerts its interactions with the residues outside of the pocket. The interaction with the residue in the pocket should contribute more to the antagonistic effect than that with the residue out of the pocket. All modeling results shown above highly indicate that acarbose might have a lesser antagonistic effect than hyperoside due to the difference in the binding mode. However, to generate the crystal structure of the α-glucosidase from *Saccharomyces cerevisiae* complexed with hyperoside or acarbose is needed to further establish the nature of their respective interaction.

## 4. Conclusions

Various solvent extracts of *C. pinnatifida* and its isolated compounds were investigated with different antioxidant systems and anti-α-glucosidase activity assay. Ethyl acetate extract from *C. pinnatifida* showed significant antioxidant activities through DPPH, ABTS, superoxide, hydroxyl radical scavenging tests, and FRAP assay. TPC in the extract of ethyl acetate also showed the highest value among all solvent extracts, indicating that the ethyl acetate extract possessed the largest amounts of polyphenols, and it corresponded to the results of antioxidant activities. The acetone extract possessed relatively high ferric reducing antioxidant power among all solvent extracts. In addition, acetone extract also showed the strongest α-glucosidase inhibitory property.

Four isolated compounds from *C. pinnatifida* were quantified by HPLC and identified as epicatechin, chlorogenic acid, hyperoside, and procyanidin B2. The bioactivity assays demonstrated that all isolates displayed antioxidant activities. In addition, hyperoside showed strong anti-α-glucosidase property. As the result of molecular modeling docking, hyperoside also exhibited the high affinity with α-glucosidase and, thus, displayed potential inhibitory activity against α-glucosidase.

The comparative evaluation of antioxidant and anti-α-glucosidase activities in various solvent extracts and bioactive compounds from the fruits of *C. pinnatifida* is first mentioned in this study. The results are enough to support the importance of the suitable solvents to extract bioactive compounds. The bioactive extracts and their isolated compounds mentioned above can be used as natural antioxidants in the food industry and dietary supplements against oxidative damage. Furthermore, the acetone extract and hyperoside can also be used as natural α-glucosidase inhibitors.

## Figures and Tables

**Figure 1 antioxidants-11-00320-f001:**
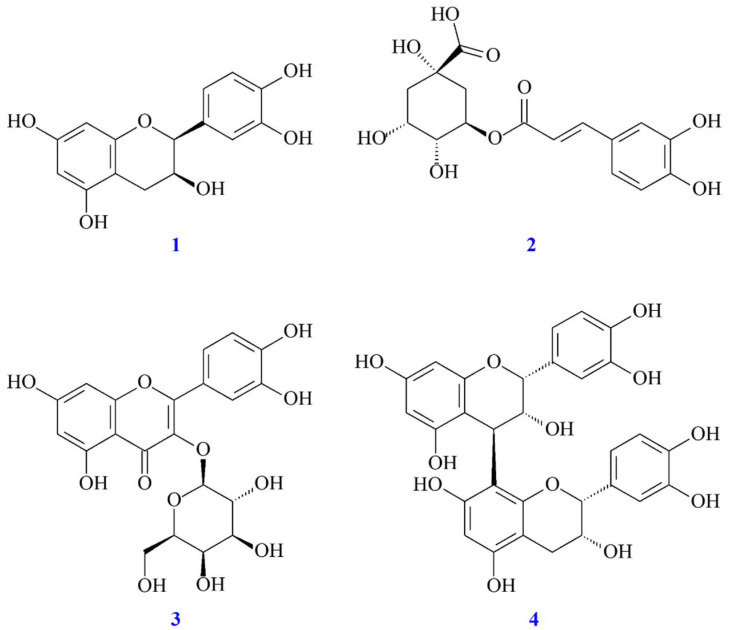
Chemical structures of epicatechin (**1**), chlorogenic acid (**2**), hyperoside (**3**), and procyanidin B2 (**4**) from *C. pinnatifida*.

**Figure 2 antioxidants-11-00320-f002:**
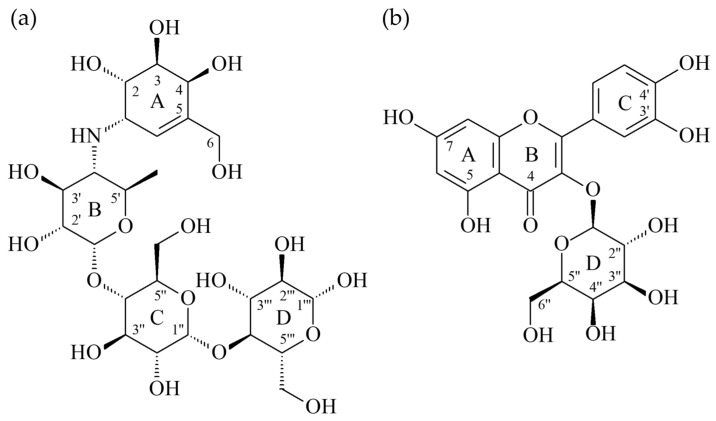
The structures of acarbose (**a**) and hyperoside (**b**).

**Figure 3 antioxidants-11-00320-f003:**
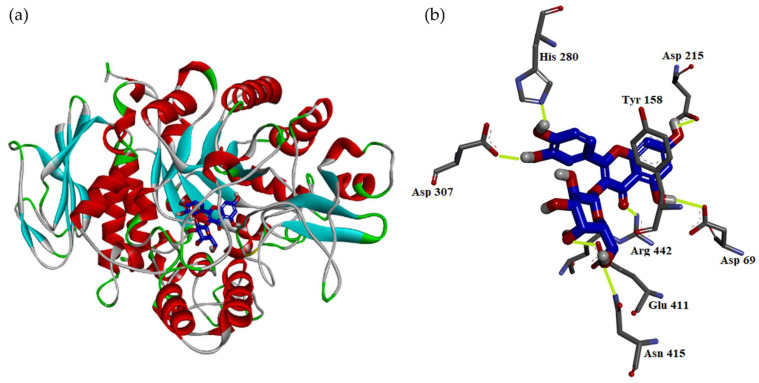
Interaction of hyperoside with active sites of *S. cerevisiae* α-glucosidase. The docking model between hyperoside and α-glucosidase (**a**). The hydrophilic binding mode between hyperoside and α-glucosidase (**b**).

**Table 1 antioxidants-11-00320-t001:** TPC, TFC, and extraction yields of *Crataegus pinnatifida* with each extraction solvent.

Extracting Solvents	Relative Polarity	TPC (mg/g) ^a^ (GAE)	TFC (mg/g) ^b^ (QE)	Yields (%) ^c^
*n*-Hexane	0.009	12.12 ± 0.26 *	31.74 ± 1.07 *	2.3 ± 1.67
Chloroform	0.259	21.09 ± 0.99 **	37.93 ± 5.31 *	3.2 ± 0.79
Dichloromethane	0.269	19.77 ± 0.13 **	34.87 ± 1.95 *	9.7 ± 1.49
Ethyl acetate	0.228	63.53 ± 0.27 ***	25.22 ± 4.21 *	15.5 ± 1.13
Acetone	0.355	45.81 ± 0.61 ***	19.41 ± 3.00 *	17.1 ± 0.23
Ethanol	0.654	32.82 ± 1.21 ***	22.61 ± 0.90 *	24.1 ± 0.23
Methanol	0.762	44.20 ± 1.30 ***	17.40 ± 3.11	28.0 ± 0.75

^a^ TPC was expressed in mg of gallic acid equivalents (GAE) per gram of extract. ^b^ TFC was expressed in milligram of quercetin equivalents (QE) per gram of extract. Values are expressed as means ± standard error. ^c^ Yield was calculated as % yield = (weight of extract/initial weight of dry sample) × 100; * *p* < 0.05, ** *p* < 0.01, *** *p* < 0.001 compared with the control.

**Table 2 antioxidants-11-00320-t002:** The antioxidant activities of different solvent extracts from *Crataegus pinnatifida* determined by DPPH, ABTS, superoxide, and hydroxyl radical scavenging and FRAP assays.

Extracting Solvents	DPPH IC_50_ (μg/mL)	ABTS IC_50_ (μg/mL)	Superoxide IC_50_ (μg/mL)	Hydroxyl IC_50_ (μg/mL)	FRAP (mM/g) (TE) ^b^
*n*-Hexane	199.18 ± 16.82 *	131.14 ± 7.15 *	>400	124.45 ± 6.27 *	70.11 ± 4.13 *
Chloroform	68.61 ± 3.14 *	141.97 ± 3.70 *	>400	114.41 ± 1.80 *	149.98 ± 5.54 *
Dichloromethane	76.22 ± 4.90 *	131.44 ± 7.13 *	>400	>400	154.05 ± 4.62 **
Ethyl acetate	23.26 ± 1.97 **	50.73 ± 8.03 *	122.95 ± 9.07 *	92.30 ± 7.47 *	216.32 ± 9.13 *
Acetone	40.06 ± 0.18 **	87.75 ± 2.70 **	>400	127.25 ± 2.26 *	264.27 ± 12.11 *
Ethanol	96.21 ± 4.26 **	140.29 ± 6.76 *	>400	139.03 ± 17.33 *	227.59 ± 10.13 *
Methanol	74.13 ± 5.61 **	105.86 ± 6.25 *	242.95 ± 18.51	138.04 ± 21.91 *	198.94 ± 8.24 *
BHT ^a^	34.28 ± 1.40 *	12.34 ± 0.29 *	N.A. ^c^	61.51 ± 2.46 *	4257.97 ± 44.90 **

Results are expressed as half inhibitory concentration (IC_50_) of each free-radical scavenging activity. ^a^ Butylated hydroxytoluene (BHT) used as positive control. ^b^ FRAP was expressed in millimolar (mM) of Trolox equivalents (TE) per gram of extract; ^c^ N.A. indicates not available; * *p* < 0.05, and ** *p* < 0.01 compared with the control.

**Table 3 antioxidants-11-00320-t003:** α-Glucosidase inhibitory activities of different solvent extracts.

Extracting Solvents	α-Glucosidase IC_50_ (μg/mL)
*n*-Hexane	99.75 ± 4.37 *
Chloroform	207.46 ± 9.52 *
Dichloromethane	120.41 ± 6.44 *
Ethyl acetate	126.36 ± 9.81 *
Acetone	42.35 ± 2.48 **
Ethanol	58.69 ± 6.91 *
Methanol	52.02 ± 0.24 **
Acarbose ^a^	317.80 ± 16.36 *

^a^ Acarbose used as positive control; * *p* < 0.05 and ** *p* < 0.01 compared with the control.

**Table 4 antioxidants-11-00320-t004:** Identification and quantification of the major active components from *Crataegus pinnatifida* in different solvent extracts.

Extracting Solvents	Epicatechin (μg/kg)	Chlorogenic Acid (μg/kg)	Hyperoside (μg/kg)	Procyanidin B2 (μg/kg)	Total Amount (μg/kg)
*n*-Hexane	0.45 ± 0.17	0.44 ± 0.28	0.63 ± 0.32	0.42 ± 0.24	1.92 ± 0.25
Chloroform	1.03 ± 0.16	1.64 ± 0.16	1.26 ± 0.22	0.82 ± 0.13	4.75 ± 0.17
Dichloromethane	2.63 ± 0.28	1.83 ± 0.24	1.02 ± 0.23	0.83 ± 0.03	6.31 ± 0.20
Ethyl acetate	0.86 ± 0.13	15.43 ± 0.62	1.62 ± 0.21	1.52 ± 0.18	19.43 ± 0.29
Acetone	2.84 ± 0.62	3.85 ± 0.62	4.24 ± 0.24	6.21 ± 0.36	17.14 ± 0.46
Ethanol	12.43 ± 0.52	3.85 ± 0.62	3.71 ± 0.48	13.63 ± 0.33	33.62 ± 0.49
Methanol	3.73 ± 0.42	6.27 ± 0.38	3.30 ± 0.38	14.49 ± 0.62	27.76 ± 0.45

Results are expressed as micrograms of each compound in kilogram of extract.

**Table 5 antioxidants-11-00320-t005:** The antioxidant activities of isolated components from *Crataegus pinnatifida* determined by DPPH, ABTS, superoxide, and hydroxyl radical scavenging and FRAP assays.

Compounds	DPPH IC_50_ (μg/mL)	ABTS IC_50_ (μg/mL)	Superoxide IC_50_ (μg/mL)	Hydroxyl IC_50_ (μg/mL)	FRAP (mM/g) (TE) ^b^
Hyperoside	5.29 ± 0.17 **	18.00 ± 0.16 *	10.59 ± 0.22 **	N.A.	7726.67 ± 39.97
Epicatechin	2.53 ± 0.25 *	3.75 ± 0.03 **	41.56 ± 0.20 *	20 ± 3.49	13,165.19 ± 42.42
Chlorogenic acid	4.85 ± 0.11 **	8.68 ± 0.08 *	99.62 ± 1.862 *	18.38 ± 0.56 **	3684.74 ± 34.19
Procyanidin B2	2.81 ± 0.08 *	2.91 ± 0.57 **	37.22 ± 0.63 *	13.69 ± 1.88 **	11,092.44 ± 32.92
BHT ^a^	24.04 ± 4.40 **	16.13 ± 3.29 ***	N.A. ^c^	60.62 ± 2.58 *	4257.97 ± 44.90 **

Results are expressed as half inhibitory concentration (IC_50_) of each free-radical scavenging activity. ^a^ Butylated hydroxytoluene (BHT) used as positive control. ^b^ FRAP was expressed in mM of Trolox equivalents (TE) per gram of extract. ^c^ N.A. indicates not available; * *p* < 0.05, ** *p* < 0.01, and *** *p* < 0.001 compared with the control.

**Table 6 antioxidants-11-00320-t006:** α-Glucosidase inhibitory activities of isolated compounds.

Extracting Solvents	α-Glucosidase IC_50_ (μg/mL)
Epicatechin	81.79 ± 6.94 *
Chlorogenic acid	170.37 ± 10.68
Hyperoside	34.98 ± 0.89 **
Procyanidin B2	118.44 ± 7.34
Acarbose ^a^	317.80 ± 16.36 *

^a^ Acarbose used as positive control; * *p* < 0.05 and ** *p* < 0.01 compared with the control.

## Data Availability

Data are contained within the article.

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
