# Peer review of "Antioxidant and Anti-α-Glucosidase Activities of Various Solvent Extracts and Major Bioactive Components from the Fruits of Crataegus pinnatifida"

_antioxidants, 2022, doi:10.3390/antiox11020320_

Round 1

Reviewer 1 Report

I commend the authors for some sound work on the antioxidant and anti-α-glucosidase activities of sol-vent extracts and bioactive components from Crataegus pinnatifida. The appropriate experiments were conducted and demonstrated some potential properties. The manuscript should initiate some further interest in the industry and its market segments. I recommend some minor adjustments of the written paragraphs and a minor spell check.

Author Response

Please see an attached file.

Reviewer 2 Report

The manuscript deals with the Antioxidant and Anti-α-glucosidase Activities of Various Solvent Extracts and Major Bioactive Components from the Fruits of Crataegus pinnatifida.

Most of the data given are already known and published as can be seen from the following reviews that are not included at the references of the manuscript:

  • Tunde Jurikova, Jiri Sochor, Otakar Rop, Jiri Mlcek, Stefan Balla , Ladislav Szekeres, Vojtech Adam and Rene Kizek. Polyphenolic Profile and Biological Activity of Chinese Hawthorn (Crataegus pinnatifida BUNGE) Fruits. Molecules 2012, 17, 14490-14509; doi:10.3390/molecules171214490
  • Shahrzad Dehghani, Soghra Mehri, and Hossein Hosseinzadeh. The effects of Crataegus pinnatifida (Chinese hawthorn) on metabolic syndrome: A review. Iran J Basic Med Sci. 2019 May; 22(5): 460–468. doi: 22038/IJBMS.2019.31964.7678

The references section is very poor and do not include all relevant published scientific papers.

There are only few new data concerning the Molecular modeling docking study, but I do not believe that this is enough to support publication of the manuscript to Antioxidants.

Author Response

Please see an attached file.

Reviewer 3 Report

Overview of the manuscript
The work is focused on the analysis of antioxidant properties of the compounds extracted from the fruit of C. pinnatifida. Different extraction protocols, using polar and non-polar solvent are described in detailed in the present work.

The total phenolic and flavonoid components of the fruit have been determined as their radical and superoxide scavenging activity. Furthermore, four specific components of the C. pinnatifida fruit have been extracted and the antioxidant and anti-α-glucosidase properties have been studied

GENERAL COMMENT

The work is interesting, and the experimental plan is well performed. The analysis of the active compounds has well detailed by the use of multiple extraction methods, which clearly illustrate the relationships between the single extractive methodology and what it achieves in term of compounds amount. However, the work suffers of a widely and seriously bad presentation. Several experimental procedures should be presented in a more detailed way and more graphs should be used to explain the results.

Finally, the manuscript remains difficult to read and several points are rather obscure due to a non-correct use of English language.

SPECIFIC COMMENTS

Abstract

In general, the abstract should be rewrite entirely, with a major attention to logical sequences and English syntax

Pag. 1, line 4-6: the sentence remains confusing. Rewrite it

Pag. 1, line 9: “…decreasing post-prandial blood sugar levels…” You didn’t monitor it. Delete the indication or rephrase the sentence.

Pag. 1, line 11: “…compare various solvent extracts…” what? You compare solvent? Extract compound? Or what. The indication is repeated throughout the manuscript, causing difficult in reading. Explain better or rephrased the indication

Pag. 1, line 18: give indication about acarbose. Which is it?

Introduction

Pag. 2, line 3-4: the sentence expresses an excessive concept, the commercially approved synthetic antioxidant are not that dangerous. Delete the sentence.

Pag. 2, line 17-21: The period introduces a topic about diabetes and pancreatic cells that is not part of your aims. Delete the period.

Pag. 2, line 40-43:  Why has been evaluated only the molecular docking between hyperoside and α-glucosidase?

Materials and Methods

Pag. 3, line 14: “…extracted with shaking…” what? Compound o other. Be clearer.

Pag. 3, line 22-25: TLC results are not shown, they should be. The speech about the fractions A1-A14 remains not clear in absence of shown results.

Pag. 3, line 49-50: The sentence is unreadable. Rewrite it.

Pag. 4, line 26: Your procedure is not clear. Indicate the concentrations of the extracts. Do you mean compounds? 100 µl of extract? It is not clear. Explain better.

Pag. 4, DPPH and ABTS radical scavenging activity:  the methods should be better explained in their differences

Pag. 5, line 41: Your procedure is not clear. Indicate the concentrations of the extracts. Do you mean compounds? 100 µl of extract? It is not clear. Explain better.

Pag. 5, line 48: Who is the control?

Results and Discussion

Pag. 7, line 12-13 (3.3 ABTS etc.): the first sentence remains unreadable. Rewrite it and explain better

Pag. 8, line 2-3 (3.7 anti-α- etc.): “regulate DM” avoid reference to the pathology in results section. It is not your topic.

Pag. 10, line 26-28: why you have analysed the hyperoside interaction with α-glucosidase only? What about the other compounds extracted?

Author Response

Please see an attached file.

Round 2

Reviewer 2 Report

I cannot identify any major revisions and corrections to have been done. When I suggested the two references I used them as an example for many more existing to the field and I expected the authors to spend some time to review more carefully the existing literature.

Author Response

Please see an attached file.

Reviewer 3 Report

no more concerns

Author Response

Please see an attached file.

Round 3

Reviewer 2 Report

I can not spot any major differences and revisions from the the original version of the manuscript.

Author Response

Please see an attached file.
